# Bovine Colostrum Supplementation in Rabbit Diet Modulates Gene Expression of Cytokines, Gut–Vascular Barrier, and Red-Ox-Related Molecules in the Gut Wall

**DOI:** 10.3390/ani14050800

**Published:** 2024-03-04

**Authors:** Federica Riva, Susanna Draghi, Alessia Inglesi, Joel Filipe, Paola Cremonesi, Antonio Lavazza, Patrizia Cavadini, Daniele Vigo, Stella Agradi, Laura Menchetti, Alessia Di Giancamillo, Lucia Aidos, Silvia Clotilde Modina, Nour Elhouda Fehri, Grazia Pastorelli, Valentina Serra, Claudia Maria Balzaretti, Marta Castrica, Marco Severgnini, Gabriele Brecchia, Giulio Curone

**Affiliations:** 1Department of Veterinary Medicine and Animal Sciences, University of Milan, Via dell’Università 6, 26900 Lodi, Italy; federica.riva@unimi.it (F.R.); susanna.draghi@unimi.it (S.D.); alessia.inglesi@unimi.it (A.I.); joel.soares@unimi.it (J.F.); daniele.vigo@unimi.it (D.V.); lucia.aidos@unimi.it (L.A.); silvia.modina@unimi.it (S.C.M.); nour.fehri@unimi.it (N.E.F.); grazia.pastorelli@unimi.it (G.P.); valentina.serra@unimi.it (V.S.); claudia.balzaretti@unimi.it (C.M.B.); gabriele.brecchia@unimi.it (G.B.); giulio.curone@unimi.it (G.C.); 2Istituto di Biologia e Biotecnologia Agraria (IBBA), National Research Council (CNR), Via Einstein, 26900 Lodi, Italy; cremonesi@ibba.cnr.it; 3Virology Laboratory, Istituto Zooprofilattico Sperimentale della Lombardia e dell’Emilia Romagna (IZSLER), Via Bianchi 9, 25124 Brescia, Italy; antonio.lavazza@izsler.it (A.L.); patrizia.cavadini@izsler.it (P.C.); 4School of Biosciences and Veterinary Medicine, University of Camerino, Via Circonvallazione 93–95, 62024 Matelica, Italy; laura.menchetti@unicam.it; 5Department of Biomedical Sciences for Health, University of Milan, Via Mangiagalli 31, 20133 Milan, Italy; alessia.digiancamillo@unimi.it; 6Dipartimento di Biomedicina Comparata e Alimentazione—BCA, University of Padua, Viale dell’Università 16, 35020 Legnaro, Italy; marta.castrica@unipd.it; 7Institute of Biomedical Technologies (ITB), National Research Council (CNR), Via Fratelli Cervi 93, 20090 Segrate, Italy; marco.severgnini@itb.cnr.it

**Keywords:** nutraceutical, bovine colostrum, gene expression

## Abstract

**Simple Summary:**

In this study focused on rabbits, a species that plays crucial roles in the EU as livestock, pets, and laboratory animals, the challenge of bacterial infections has led to a search for alternatives to antibiotics. Bovine colostrum (BC), known for its content in immunoregulatory compounds, antimicrobial peptides, and growth factors, has being explored for disease treatment and prevention. Our research investigates the impact of BC diet supplementation on rabbit intestines, particularly examining gene expression. The study reveals that BC has varying effects on different genes in the jejunum, cecum, and colon, influencing inflammatory and antioxidant responses. The findings suggest a potential role for BC in modulating the rabbit gastrointestinal tract, emphasizing the need for further research to fully understand its histological and physiological impact.

**Abstract:**

Rabbits, pivotal in the EU as livestock, pets, and experimental animals, face bacterial infection challenges, prompting a quest for alternatives to curb antibiotic resistance. Bovine colostrum (BC), rich in immunoregulatory compounds, antimicrobial peptides, and growth factors, is explored for disease treatment and prevention. This study assesses BC diet supplementation effects on rabbit intestines, examining gene expression. Thirty female New Zealand White rabbits at weaning (35 days) were divided into three experimental groups: control (commercial feed), 2.5% BC, and 5% BC. The diets were administered until slaughtering (81 days). BC-upregulated genes in the jejunum included IL-8, TGF-β, and CTNN-β1 at 5% BC, while PLVAP at 2.5% BC. Antioxidant-related genes (SOD1, GSR) were downregulated in the cecum and colon with 2.5% BC. BC 5% promoted IL-8 in the jejunum, fostering inflammation and immune cell migration. It also induced genes regulating inflammatory responses (TGF-β) and gastrointestinal permeability (CTNN-β1). BC 5% enhanced antioxidant activity in the cecum and colon, but no significant impact on anti-myxo antibody production was observed. These results suggest that BC has significant effects on the rabbit gastrointestinal tract’s inflammatory and antioxidant response, but further research is required to fully understand its histological and physiological impact.

## 1. Introduction

Italy is the third rabbit meat producer within the European Union, after Spain and France, and is in the top five producers in the world [1]. Despite this extensive and continuously growing rabbit meat production, its consumption decreased in the last five years, and the main causes of this trend are an increasing association with the perception of rabbits as pets [2,3,4]. Meat rabbits are among the most farmed species in the EU, but these animals are also used as laboratory animals and kept as pets by many families [5]. Rabbit breeding presents some issues due to the sensitivity of this species to many bacterial infections causing, in particular, respiratory and intestinal diseases. This susceptibility causes high mortality of animals affecting productivity in the case of farms [6] or study results when they are used as laboratory animals [7]. The treatment of these pathologies requires the use of antibiotics, but under a One Health approach to avoid antibiotic resistance, alternative therapeutic strategies are badly needed [8,9]. Interest in the development and use of natural-based products, called nutraceuticals, has increased in the last years. Nutraceuticals are food and feed, or parts of them, containing biologically active compounds which may have a positive role in prevention and treatment of several diseases in both humans and animals [10]. Among these nutraceuticals, bovine colostrum (BC) has been shown to have a preventive and therapeutic effect on gastrointestinal diseases such as IBD, both in different animal species including humans [11,12,13,14]. Specifically, colostrum is the first secretum of the mammary glands after giving birth, and its main function consists in the supply of passive immunity from the mother to the offspring, especially in species with greater complexity of the placental barrier [15]. BC has high similarities with human colostrum and, compared with the colostrum from other ungulates, it is rich in more than 90 biologically active molecules with specific functions and properties [16]. Examples of these molecules are antimicrobial peptides (glycomacropeptide, lactoferrin, and lysozyme); immune-regulating compounds; exosomes, which contain long non-coding RNA and microRNA; and growth factors, which are absent in the mature milk [17]. All these molecules present different functions such as reinforcing the natural defense system, improving protection against pathogens, ensuring the development of the immune system and modulating the immune response, balancing intestinal microbiota, and enhancing the growth and repair of various tissues [18,19,20]. Colostrum provides both adaptive and passive immunity to the newborn. The major form of adaptive immunity in the gut is humoral immunity direct to microbes in the lumen; this function is mainly mediated by IgA antibodies secreted in the gut or, in the case of breast-feeding infants, from IgA present in colostrum. The antibodies in the lumen bind to commensals and pathogens and prevent passage through the mucosal epithelial barrier and the following colonization of other tissues [21]. Moreover, BC contains several antimicrobial and immune-stimulating factors. Examples are lactoferrin, lactoperoxidase, lysozyme, α-lactalbumin, and peptides derived from caseins such as glycomacropeptide (GMP) and whey protein; these compounds show high antimicrobial activity [12]. Oligosaccharides, gangliosides, and nucleosides are also present in BC and are able to provide protection against pathogens acting as “false receptors” of intestinal cells [22]. Another type of immune-related molecule produced in the mammary gland and released into colostrum are cytokines. Cytokines are a group of proteins and glycoproteins performing their functions at minute or small concentrations (10 to 1000 pg/mL) [23]. These molecules modulate the development of the newborn immune system and manage the inflammatory response and antibody production because of their immunomodulatory activity. The most relevant cytokines found in both colostrum and mature milk include interleukins (IL), such as IL-1β, IL-6, IL-8, IL-10, and IL-12, and tumor necrosis factor (TNF)-α [24]. Moreover, cytokines regulate the function of the mucosal barrier at different levels through their chemotactic activity, neutrophils recruitment, and IgA production stimulation. BC is widely studied also for its ability to influence gut microbiota through its prebiotic activity. In several studies, mainly conducted in humans, BC showed the ability to promote the growth of beneficial microbiota families, such as *Lactobacillaceae*, *Bifidobacteriaceae*, *Lachnospiraceae*, *Akkermansiaceae*, and *Bacteroidaceae*, which in turn increase the strength of immune defenses and modulate the physiology of the digestive system [25,26,27]. This capacity is due to the fact that, like other prebiotics, it acts as a selective food source for certain bacterial strains and can also manifest biological activity directly on the barrier. Examples of bioactive compounds within colostrum which promote the increase of beneficial bacteria are oligosaccharides; meanwhile, immunoglobulins, lactoferrin, and cytokines resist digestion and perform their function along the gut increasing intestinal permeability [28,29,30]. The ability of BC to modulate the intestinal microbiota is of fundamental importance because it is widely recognized that gut microbiota dysbiosis, or pathogen infections, lead to disruption of both the epithelial and vascular barriers. The damage increases gut–vascular barrier (GVB) permeability, allowing, subsequently, microbes and microbe-derived products to cross the barriers and reach the systemic circulation causing an inflammatory state with systemic consequences [31]. Thanks to the mentioned features and peculiarities, bovine colostrum could serve as an excellent nutraceutical substance to be added to the diet of rabbits, aiming to strengthen their immune defenses, particularly in the intestinal compartment, to address gastro-enteric pathologies, which are one of the weak points in rabbit farming [32].

Given the important role of BC on the physiology and immunology of the gut, this study aimed to investigate the effect of a BC-supplemented diet on the immune response, permeability, and red-ox activity of different tracts of the rabbit gut. Moreover, the effect of BC supplementation on the possible synergism with vaccination against myxomatosis in terms of specific antibody production in rabbit serum was investigated. The use of an animal product in an herbivorous species is not precluded when it is a candidate for improvement of the welfare and production quality and quantity of the animals. Moreover, the aim of our study was to use rabbits as a translational model for humans, which already makes use of BC as a nutraceutical with several benefits whose molecular mechanism is still unknown.

## 2. Materials and Methods

### 2.1. Animals and Sample Collection

The experimental trial was conducted in the facilities of the Department of Agricultural, Food and Environmental Sciences of the University of Perugia, Italy. In accordance with European and Italian laws (EU Directive 2010/63 and Decreto Legislativo 26/2014) regarding laboratory animal protection, the rabbits were maintained under the supervision of a responsible veterinarian, and the experimental protocol was approved by the Ethical Committee of the Department of Veterinary Medicine of the University of Milano with the code OPBA_42_2021. All procedures were made to minimize animal discomfort and reduce the number of experimental animals. The animals used in the trial were suckled by mothers who were fed the same diet they received after weaning (35 days) till 81 days of age. The rabbits were also vaccinated against myxomatosis at 35 days of life. According to dietary treatment, 30 New Zealand White female rabbits were randomly assigned to three groups: the control group (n = 10 animals, CTRL) was given standard feed, while another group was provided feed enriched with 2.5% bovine colostrum (n = 10, 2.5% BC group) and a third group received feed supplemented with 5% bovine colostrum (n = 10, 5% BC group). The BC was collected from the first milking of healthy cows (no mastitis, laminitis, endometritis, or metabolic disorders observed) under the supervision of expert veterinarians. More specifically, the colostrum was obtained from multiparous Friesian cows raised on a single farm, and only high-quality colostrum (Brix grade > 22%) was used. The quality was tested using a digital refractometer (Palm Abbe, Misco, Solon, OH, USA). Immediately after the quality assessment, the bovine colostrum was pooled and frozen. Before the start of the experiment, the basic composition and IgG concentration of BC were evaluated: total solids 22.1%, fat 4.5%, total protein 14.2%, lactose 2.9%, and IgG 3.3 g/100 mL. Then it was lyophilized and added to the feed of the treated groups before pelleting. Both the chemical composition of the diets and the diet formulation have been previously described [33].

Rabbits were raised in individual cages (600 × 250 × 330 mm) after weaning and kept in a conditioned environment with temperature ranging from 18 to 20 °C, relative humidity of 60–65%, and a photoperiod of 16 h of light for the duration of the entire trial; water and feed were provided ad libitum [34]. Rabbits were slaughtered at an official slaughterhouse after stunning by electronarcosis, and the carcasses were immediately chilled (4 °C) until the analysis time. At slaughter, the digestive tract was removed from each animal; tissue samples of each section (jejunum, caecum, and colon) and a mesenteric lymph node were collected with the use of a sterile scalpel and stored in 2 mL sterile tubes in RNAlater^®^ (Sigma-Aldrich, St. Louis, MO, USA). Samples were stored at −80 °C until processing, as previously described [35]. The content of each intestinal tract was collected in a 2 mL sterile tube and immediately frozen at −20 °C until use. Moreover, blood samples were collected at different time points from the marginal ear vein, without anticoagulant. After an incubation of 15 min at room temperature, samples were centrifuged and the serum was transferred to a new sterile tube and stored at −20 °C until use.

### 2.2. RNA Extraction, Reverse Transcription, and Real-Time PCR from Rabbit Tissues

Total RNA was isolated from jejunum, caecum, colon, and lymph node samples using TRIreagent (Sigma-Aldrich, St. Louis, MO, USA), according to the manufacturer’s protocol. After the extraction, the RNA concentration was determined using a spectrophotometer (BioPhotometer, Eppendorf, Hamburg, Germany) at 260 nm wavelength. Two µg of the total RNA from each sample was retrotranscribed using a High Capacity cDNA Reverse Transcription Kit (Applied Biosystem, Foster City, CA, USA), according to the manufacturer’s instructions. The obtained cDNA was used as a template for real-time PCR in 25 µL optimized reaction volume using Sybr Green (Applied Biosystem, Foster City, CA, USA), as previously described [28]. The primers were designed using Primer Express Software (Applied Biosystem, Foster City, CA, USA) and purchased from Eurofins (Luxembourg City, Luxembourg). Their sequences are shown in Table 1. The rabbit actin β (ACTβ) gene, amplified by specific primers, was used as housekeeping.

IL-8, IL-10, TGF-1, PLVAP, CTNN-β1, SOD1, and GSR genes were investigated in the jejunum, caecum, and colon; at the same time, CSF3R, IL-8, IL-10, and TGF-β were investigated in mesenteric lymph nodes. Each sample was analyzed in duplicate, and a no-template control (NTC) for each gene was included in each plate. Real-time quantitative PCR was carried out in the Applied Biosystems™ QuantStudio™ 5 Real-Time PCR System (Applied Biosystem, Foster City, CA, USA). The expression of rabbit target genes was normalized using the calculated ACT-β cDNA expression of the same sample and run. We used a single housekeeping gene (actin b) both for technical reasons (to have numerous primer pairs for the same sample on the same plate) and because that gene was used in several previous publications on rabbits [36,37], but we are aware that this approach could be a bias. Indeed, the MIQE guidelines suggest to use at least three housekeeping genes to normalize the expression of target genes. The relative quantification of each gene was calculated through the method of “delta Ct (DCt)”, as described by Schmittgen and Livak (2008) [38]. Results are expressed as arbitrary units (AU), obtained by a multiplication by “10,000” of the 2^−DCt^ value.

### 2.3. Gram Micro-Organism Staining

Liver samples were collected and fixed in buffered 10% formalin (Bio-Optica, Milan, Italy) for 24 h, dehydrated, and embedded in paraffin. Microtome sections (4 μm thickness) of the tissue were analyzed for the detection of bacteria, as described by Tripathi et al. 2023 [39]. Briefly, sections were stained with a crystal violet solution and fixed with iodine; subsequently, they were treated with acetone and counterstained with safranin. Bacteria were stained blue and cell nuclei were stained red. On these sections, a blind count of the bacteria was performed on six hepatic central veins for each animal.

### 2.4. Microbiota Analysis

Microbial profiles based on 16S rRNA amplicon-based sequencing for all the animals, intestinal tracts, and diets have already been presented elsewhere (Agradi et al., 2023 [35]). Briefly, the bacterial DNA was extracted using a QIAamp PowerFecal Pro DNA Kit (Qiagen, Hilden, Germany), amplified following the 16S Metagenomic Sequencing Library Preparation Protocol (Illumina, San Diego, CA, USA), and sequenced on a MiSeq (Illumina) instrument with 2 × 250-base paired-end reads run. Data analysis involved a preliminary filtering and trimming, clustering into zero-radius operational taxonomic units (zOTUs), and taxonomic classification against the SILVA 138 database using 0.5 as the confidence threshold using PandaSeq, QIIME 1.9.0 suite, USEARCH (v. 11.0.667), and RDP classifier. Bacterial composition was expressed as relative abundances of taxa at family and genus levels. Functional predictions of the microbial profiles were performed in PICRUSt2 (v2.5.1). The data presented in this study are openly available in the NCBI Short Read Archive (SRA) under experiment IDs SRR22879769–SRR22879893 (BioProject ID PRJNA915237, https://www.ncbi.nlm.nih.gov/bioproject/PRJNA915237, accessed on 5 January 2023).

### 2.5. Vaccination Efficacy Evaluation

Rabbits were vaccinated against myxomatosis disease with the vaccine Cunivax Myxoma (Fatro Industria Farmaceutica Veterinaria S.p.A., Ozzano Emilia (BO), Italy) at 35 days of age. Blood samples were collected at T0 (the day of vaccination for the pre-immune serum) and at T1 (28 days post-vaccination) from the marginal ear vein, without anticoagulant. Samples were centrifugated for 15 min at 3000× *g*, and the serum was transferred into a new 1.5 mL sterile tube and stored at −20 °C until use. Antibody anti-myxo titer was measured using the cELISA method [40]. cELISA for myxomatosis was developed and validated in-house by IZSLER. The method is based on liquid-phase competition for antigen (m71L membrane protein) between antibodies from test serum and AcM 1E5 absorbed into the wells of the plate. Any anti-myxo virus antibodies present in the serum, binding to the m71L protein, prevents/reduces its binding to the solid phase. The resulting quantitative decrease in protein is detected in the final ELISA step by the addition of the same peroxidase, labeled AcM 1E5. According to the test procedures applied, a serum is considered negative when the absorbance value A492 of the first dilution (1/10) decreases by less than 15% of the reference value (negative serum 1/10), while it is positive when it decreases by more than 25%. The titer of positive sera corresponds to the dilution of the serum which inhibits the A492 value of negative serum of 50% ± 20.

### 2.6. Statistical Analysis

Statistical analyses were performed using GraphPad Prism 6 (La Jolla, CA, USA) considering statistically significant values at *p* < 0.05 and tendencies at *p* < 0.1. Shapiro–Wilk tests and diagnostic graphics were used to verify the distribution of data and the equality of variances. The expression of target genes as well as bacterial counts of the central veins and Ig titer against myxomatosis among the experimental groups (control, 2.5%, and 5% BC supplementation) were compared by ANOVA or the Krustal–Wallis test depending on the normal or not-normal distribution of the data. The Mann–Whitney test was used to analyze the differences between two treatments in the same kind of sample. The data were presented as least-squared-means SEM (standard error of the mean).

Finally, correlations between the bacterial composition, functional predictions, and gene expression of the measured genes for the intestinal tracts were estimated by Spearman’s rank correlation. Bacterial composition was evaluated at the genus or family level, according to the lowest level for which a classification was possible; pathway abundances were considered at the L4 level. In order to limit the number of correlations to the most meaningful ones, only families and genera present at >1% on average and L4 pathways reported as differential in Agradi et al., 2023 [35] were considered. Since the original publication included microbial profiles and functional predictions for both the luminal and mucosal bacteria, in the present paper, only the latter (i.e., mucosa) were considered to be more likely associated with the expression values of inflammatory- or GVB permeability-related genes. A *p*-value < 0.05 on the linear model was considered significant.

The effect of the group on the body weight (BW) of rabbits at slaughtering was evaluated by univariate analysis of variance including BW at weaning as covariates. Sidak correction was used for multiple comparisons.

## 3. Results

### 3.1. BC Supplementation Modulates Immune-Related Gene Expression in the Small Intestine

Given that BC is implicated in the regulation of immune cell recruitment and inflammatory response, we investigated the gene expression of some cytokines involved in these activities in the gut wall samples and in mesenteric lymph nodes. In our model, BC supplementation modulates the expression of IL-8 and TGF-β in the small intestine (jejunum), but not in the large intestine (caecum and colon). In particular, we observed a dose-dependent increased expression of IL-8 (*p* = 0.0037) and TGF-β (*p* = 0.0075) in the jejunum of rabbits fed with the BC-supplemented diet as compared to the control group (Figure 1). On the contrary, IL-10 expression was not different among the three experimental groups neither in the small nor in the large intestine (Figure 1).

### 3.2. BC Supplementation Modulates Gut–Vascular Barrier-Related Gene Expression in the Gut

The gut–vascular barrier (GVB) represents an important anatomical structure for the maintenance of the gut and whole organism homeostasis. We analyzed the gene expression of PLVAP and CTNN-β1, two important markers of the integrity of the GVB. Interestingly, supplementation with 5% of BC induced a significant increase of the CTNN-β1 expression in the jejunum (*p* = 0.0297) and a modulation of PLVAP compared to the control group (Figure 2). The supplementation of 2.5% of BC, on the other hand, did not induce any modulation of CTNN-β1 but induced a slight significant increase of PLVAP (*p* = 0.0307) as compared to the control group (Figure 2). In the caecum, we observed a similar trend (not significant) with an increase in CTNN-β1 (5% BC supplementation, *p* = 0.0954) and PLVAP (2.5% BC supplementation) compared to the control group (Figure 2). In the colon, BC supplementation did not modulate the gene expression of CTNN-β1 and PLVAP (Figure 2).

Given the modulation of the genes involved in the integrity of the GVB after BC supplementation in the diet of rabbits, we investigated the presence of bacteria that could reach the liver from the intestine via portal veins. The presence of bacteria in the liver was evident in the lumen of the central veins and rarely in the hepatic tissue in all groups (Figure 3A–C, arrows). The number of bacteria was significantly higher in rabbits fed the experimental diets containing bovine colostrum when compared with the control group (Figure 3; BC 5% > CTR, *p* < 0.01; BC 2.5% > CTR, *p* < 0.05).

### 3.3. BC Supplementation Modulates Antioxidant-Related Gene Expression in the Gut

Research on nutraceutical products has highlighted the importance of antioxidant activity to promote homeostasis not only of the gut, but also of the whole organism. In this view, we investigated the role of BC on the modulation of the expression of SOD1 and GRS genes in the gut. In the jejunum, both genes were not modulated in the BC (2.5% or 5%) groups compared to the control group. BC supplementation (2.5%) significantly downregulated SOD1 both in the caecum (*p* = 0.0087) and the colon (*p* = 0.0158) (Figure 4). BC (5%) did not impact the expression level of SOD1 in the large intestine (caecum and colon). GSR gene expression was not modulated in the gut after BC supplementation of the diet compared to the control group (Figure 4).

### 3.4. BC Supplementation Does Not Modulate Immune-Related Genes in Lymph Nodes

We also investigated if BC supplementation could affect mucosal immunity in terms of gene expression modulation in mesenteric lymph nodes. We analyzed the gene expression of IL-8, IL-10, TGF-β, and CSF3R in the mesenteric lymph nodes of the rabbits of the three experimental groups, but we did not observe any modulating effect of BC supplementation compared to the control group (Figure 5).

### 3.5. Response to Vaccination

In order to evaluate the possibility that BC supplementation acts synergistically with anti-viral myxomatosis vaccination by favoring the yield of higher Ig titers, we vaccinated the rabbits against myxomatosis at 35 days of age and measured the specific anti-myxomatosis Ig in the serum after 28 days post-vaccination. The average anti-myxo Ig did not present any statistical difference among the groups (Figure 6).

### 3.6. The Gene Expression of the Molecules under Study Correlates with the Abundance of Microbial Families

Regarding the inflammation-related genes, in jejunum samples, the control diet showed a positive correlation between *Atopobiaceae* and IL-8 and a negative one between IL-10 and several taxa (i.e., *Lachnospiraceae*, *Akkermansia*, *Bacteroides*, *Methanobrevibacter*, *Alistipes*, and *Barnesiellaceae*) (Figure 7). Supplementation of 2.5% BC incurred a positive correlation between IL-8 and *Ruminococcus*, whereas, with a 5% BC diet, *Dubosiella* was positively correlated with IL-8 and IL-10, *Metanobrevibacter* was positively correlated with IL-10, and *Muribaculaceae* and *Rikenellaceae* were negatively correlated with TGF-β (Figure 7). In the caecum, no correlation was observed for the control diet, whereas at 2.5% BC, *Lachnospiraceae* and *Ruminococcus* were positively correlated with IL-8 and *Akkermansia* with TGF-β; *Muribaculaceae* and *Atopobiaceae*, here, were negatively correlated with TGF-β expression (Figure 7). Finally, at 5% BC, *Methanobrevibacter* was inversely correlated with IL-10 expression. In colon samples, with the control diet, we reported a positive correlation between *Eggerthellaceae* and IL-8 and between *Barnesiellaceae* and *Rikenellaceae* and TGF-β; at 5% BC, *Marvinbryantia* genus was positively correlated with IL-10 (Figure 7). Concerning gut–vascular barrier permeability-related genes (i.e., CTNN-β1 and PLVAP), in jejunum samples, CTNN-β1 was correlated inversely with Bacteroides, Alistipes, *Barnesiellaceae*, and *Ruminococcus* (control diet); inversely with *Oscillospiraceae* and Rikenellaceae (5% BC); and directly with Bacteroides (2.5% BC) and *Atopobiaceae* (5% BC). In caecum samples, *Dubosiella* (control diet) and *Eggerthellaceae* (5% BC) were negatively correlated with PLVAP, whereas Bacteroides and *Barnesiellaceae* were positively correlated with CTNN-β1 with 2.5% BC. No significant correlations were reported for samples subjected to the control diet and in all colon samples (for all the three diet groups). As far as antioxidant capability-related genes are concerned, in jejunum samples, with the control diet, *Ruminococcus* and *Marvinbryantia* were positively correlated with SOD1 expression, as well as *Eubacteriaceae* with GSR; at 5% BC, *Atopobiaceae* was positively correlated with both SOD1 and GSR, whereas *Rikenellaceae* and *Oscillospiraceae* were negatively correlated with GSR. In the caecum, samples showed a direct correlation between *Lachnospiraceae*, *Ruminococcaceae*, *Barnesiellaceae* (control diet) and Bacteroides, *Barnesiellaceae* (5% BC), and SOD1. In colon samples, only one negative correlation between GSR and *Akkermansia* was reported.

### 3.7. The Gene Expression of Immune, Gut–Vascular Barrier, and Antioxidant Molecule Correlates with Specific Bacterial Pathways

In jejunum tissue, IL-8 was positively correlated with lactose degradation (in rabbits subjected to the control diet), with acetylmuramoyl-pentapeptide biosynthesis (2.5% BC integration), and with peptidoglycan biosynthesis (5% BC, together with TGF-β expression), whereas IL-10 expression was positively correlated with glutamate degradation and negatively associated with several biosynthesis (i.e., biotin, glycogen, vitamin B6, and stearate) and degradation (glutamate, histidine) pathways with the control diet and with the biosynthesis of acetylmuramoyl-pentapeptide, AIR, CDP-diacylglycerol, phosphatidylglycerol, and amino acids (phenylalanine, threonine, and tyrosine) with 5% BC supplementation (Figure 8). In caecum tissue samples, no evidence was reported for the control diet, whereas 2.5% BC showed a direct correlation for glycogen and biosynthesis (with IL-8), DHNA, and phylloquinone biosynthesis (TGF-β) and an inverse one for lactose and lysine degradation pathways (TGF-β); 5% BC supplementation resulted in a direct correlation of DHNA and phylloquinone biosynthesis (IL-10) and of biosynthesis of biotin, stearate, and vitamin B6 (TGF-β); at the same time, a negative association was reported for diterpenoid and menaquinone synthesis (IL-10) and lysine degradation (TGF-β) (Figure 8). Finally, in colon samples, DHNA, phylloquinone, and demethylmanaquinone biosynthesis (control diet), AIR biosynthesis (2.5% BC), and phenylalanine and tyrosine synthesis (5% BC) were positively related to IL-8, whereas GGPP and glycogen biosynthesis were directly associated with IL-10 in 2.5% BC and the lactose degradation pathway was negatively correlated with TGF-β with the control diet. Concerning gut–vascular barrier permeability, in jejunum samples there was a positive association for lactose degradation (control diet), stearate biosynthesis (2.5% BC), and peptidoglycan biosynthesis (5% BC) with CTNN-β1, as well as a negative one with GGPP, biotin and stearate biosynthesis, and glutamate degradation (control diet). On the other hand, PLVAP showed a direct correlation with isoprenoids, NAD and unsaturated fatty acids biosynthesis, glycogen degradation (control diet), and diterpenoid synthesis (5% BC) and an inverse one with diterpenoid synthesis (control diet) and phenylalanine and tyrosine synthesis (2.5% BC) (Figure 8). In caecum samples, we reported a positive correlation of the glutamate degradation pathway with both CTNN-β1 and PLVAP and for peptidoglycan biosynthesis with CTNN-β1 for the control diet. At higher BC contents, only one negative correlation between AIR biosynthesis and CTNN-Β1 was reported (Figure 8). Finally, in the colon, several pathways were significantly correlated, such as, in the control diet, phenylalanine and tyrosine synthesis (positive correlation with CTNN-β1); DHNA, menaquinone, phenylalanine, phylloquinone and tyrosine synthesis (positive correlation to PLVAP); diterpenoid synthesis (negative correlation with CTNN-β1); and purine nucleotides salvage (negative correlation with PLVAP). With 5% BC supplementation, diterpenoid, menaquinone, phylloquinone, and vitamin B6 synthesis were all negatively correlated with PLVAP gene expression (Figure 8). Finally, for antioxidant capability, in jejunum samples, lactose degradation (with the control diet) and the sugar nucleotides pathway (with 2.5% BC) were positively correlated with GSR, whereas CDP-diacylglycerol biosynthesis, phenylalanine, and tyrosine synthesis had the opposite behavior. SOD1 gene expression, on the other hand, was inversely correlated with CDP-diacylglycerol biosynthesis (for 2.5% BC) and directly with peptidoglycan biosynthesis and glutamate degradation (in 5% BC). In caecum tissue, several pathways, such as demethylmanaquinone and GGPP biosynthesis, glutamate degradation, isoprenoids, sugar nucleotides (control diet), glycogen degradation (2.5% BC), and stearate and biotin synthesis (in 5% BC), were positively correlated with SOD1 expression, while phenylalanine and tyrosine synthesis (control diet) were negatively correlated. GSR gene expression, on the other hand, was found to be related in a direct way to glutamate degradation (2.5% BC) and to stearate and biotin synthesis (5% BC). Finally, in the colon, phenylalanine and tyrosine synthesis were positively correlated with SOD1 for the control diet, whereas AIR biosynthesis (2.5% BC) had a negative correlation with the same gene. Vitamin B6 (control diet) was positively related and phylloquinone synthesis (5% BC) was negatively related to GSR.

### 3.8. Body Weight Did Not Differ among the Three Groups

Body weight at slaughter, corrected for BW at weaning, did not differ between groups. The estimated means of BW at slaughtering were 2.55 ± 0.04, 2.49 ± 0.04, and 2.62 ± 0.05 kg for the control, 2.5% BC, and 5% BC, respectively (*p* = 0.121).

## 4. Discussion

Within the field of animal husbandry, various nutraceutical products have been explored as substitutes for antibiotics, seeking to mitigate and diminish the growing concern of antibiotic resistance, while simultaneously fortifying the immune system. [34,41,42]. In this regard, we studied for the first time the effect of BC supplementation in the diet of weaned rabbits and its potential role on the regulation of gut immunity, gut microbiota, and potentiation of the vaccination. We demonstrated that BC affects the gene expression of immune-related genes only in the wall of the jejunum. In particular, in our model, 5% BC supplementation significantly increased IL-8 and TGF-β expression in the jejunum compared to the control and 2.5% BC groups, but no effect was observed in the large intestine (caecum and colon). The upregulation of these cytokines could be promoted by a direct effect of the bioactive components of the BC or indirectly by the modulation of the microbiota or both [35]. TGF-β is abundant in the gut and it is produced by several types of cells in response to cytokine stimulation, apoptotic cells, gut microorganisms, or through autocrine stimulation [43,44,45]. In our study, we observed a significant increase in *Clostridia* (both in the 2.5% and 5% BC supplementation), which are known to produce butyrate as a main metabolite (in particular, *Lachnospiraceae*) [46,47]. Butyrate, in turn, has been demonstrated to stimulate the expression and activation of TGF-β in the gut and the following differentiation of Treg [48,49,50]. In the gut, Treg cells play a pivotal role in the maintenance of homeostasis in the presence of a huge amount of proinflammatory stimuli deriving from microbiota and food antigens [51]. In this view, BC could favor the homeostasis of the gut by promoting the differentiation of Treg cells. In contrast, IL-8 is a proinflammatory chemokine that can be produced by macrophages and epithelial cells in response to microbial stimuli and acts by recruiting immune cells from the circulation to the tissues [52,53,54]. The IL-8 upregulation in the jejunum showed by 5% BC-supplemented rabbits could represent the response to an alteration of the microbiota in this tract or to the proinflammatory molecules (ROS) contained in the BC [55,56]. The upregulation of IL-8 and TGF-β was not evident in the large intestine (caecum and colon), notwithstanding that in the colon lumen and mucosa microbiota we observed a significant increase of *Clostridia*. This could be due to the increase of different species of *Clostridia* in jejunum compared to large intestine tracts or to the degradation along the gut of the bioactive molecules of BC stimulating that cytokine modulation.

Our results showed that BC supplementation modulates the gut vascular barrier (GVB), mainly in the jejunum and caecum (only a tendency) but not in the colon. In the jejunum and somewhat in the caecum, 5% BC supplementation increased CTNN-β1 compared to the control group. B-catenin is fundamental for the homeostasis of the gut, in particular for the renewal of the gut epithelium and for the activity of the gut vascular barrier [57,58]. This suggests that 5% BC supplementation could help the homeostasis of the gut also through the Wnt/b-catenin signaling pathway, according to the results shown by Aidos et al. 2023 [59]. To evaluate intestinal inflammation, it is well established that the enteric nervous system can be used as an indicator of intestinal activation via glial cells [60]. In a recent study of our group, we investigated the effect of bovine colostrum in rabbit gut health: an evaluation of zonulin expression, a junctional protein whose function is to prevent leakage of solutes and water and seals between the epithelial cells, showed that it was higher in the BC 5% group, suggesting increased permeability, which was partially confirmed by the increased expression of GFAP (glial fibrillary acidic protein), a marker of intestinal glial cell, but this result was limited to the duodenum [59].

Interestingly, our results differ from those obtained in the study by Rzeszotek et al., where the effects of colostrum administration in rabbit diets were assessed at different time points (1, 3, and 6 months). Surprisingly, in their study, after 3 months of supplementation, b-catenin expression in treated animals was significantly lower compared to the control animals [61]. This difference in outcomes is likely attributable to the varying daily dose of colostrum administered. Our study also shows an increase of PLVAP/PV1 in the jejunum and slightly in the caecum of the 2.5% BC-supplemented group. This suggests an increase blood endothelial cell permeability that could impair the function of the GVB and the translocation of bacteria and their metabolites into the circulation and the following dissemination to other organs [57]. According to these last results, we observed the presence of bacteria in the central veins of the liver. They were in a statistically significantly higher number in the BC 5% group; these findings favor the hypothesis that when intestinal permeability is enhanced, the clinical consequence is an increased bacterial translocation that may eventually worsen liver function [62]. Upregulation of CTNN-β1 and PLVAP was not evident in the colon and was not significant in the caecum, suggesting degradation along the gut of the bioactive molecules of BC stimulating their modulation or indirectly by a different impact of BC supplementation on the microbiota composition.

BC supplementation influences antioxidant gene-related expression in the large intestine (caecum and colon) but not in the jejunum. The downregulation of SOD1 in the caecums of the 2.5% BC-supplemented group could lead to ROS accumulation and inflammation. Indeed, SOD1 deficiency has been associated with several pathological conditions not only in the gut but also at the brain level [63]. Downregulation could be due to epigenetic mechanisms [35]. SOD1 could be also modulated indirectly by the modulation of the microbiota induced by 2.5% BC supplementation rather than directly by some element contained in the BC, given that 5% supplementation did not induce the same or greater downregulation. An alternative explanation might be linked to the abundance of antioxidant compounds present in colostrum, encompassing both enzymatic elements like lactoperoxidase, catalase, superoxide dismutase, and glutathione peroxidase, and non-enzymatic components including vitamins E, A, and C lactoferrin, and selenium [56]. Consequently, this richness could result in a diminished expression of these substances within the gastrointestinal tract. Our results contrast with what has been reported in other studies on different animal species. Specifically, in chickens, supplementation of the diet with short-term feeding of defatted bovine colostrum has been shown not only to decrease levels of intestinal inflammation but also to reduce nitro-oxidative stress [64]. A similar result regarding the reduction of oxidative stress risk has also been achieved in an intestinal ischemia/reperfusion injured rat model. Kwon et al. showed that the administration of bovine colostrum after ischemic damage increased antioxidant enzyme levels, decreased lipid peroxidation, and also mitigated cytokine effects by reducing proinflammatory enzyme levels [61].

The effect of BC seems to be local and restricted to the gut wall, since we did not observe any modulation of cytokines and growth factor receptors in the mesenteric lymph nodes. This is also supported by the fact that BC supplementation did not affect the production of anti-Mixo Ig after vaccination of the BC-supplemented rabbits, suggesting that BC supplementation does not increase the priming of B-cell response to this vaccination. The hypothesis of a specific local effect is in agreement also with the results of Serra et al. 2023, who did not see any antioxidant effect of BC supplementation in blood and liver [33]. Besides a direct effect of BC present in the supplemented diets thanks to the presence of a high content of cytokines, Ig, and growth factors, the mechanism involved in gene expression modulation could also derive from modulation of the gut microbiota that in turn affects gene expression in the gut wall. Indeed, the correlation between gene expression and relative abundance of bacterial families showed significant positive correlation among the differentially expressed genes (IL-8, TGF-β, and b-catenin) both in the control and much more in the 5% BC-supplemented groups and *Atopobiaceae* family. Interestingly, this group of bacteria has been described to be useful as a probiotic in humans and animals [65,66,67]. Moreover, the experimental group showing the highest number of significant correlations (both positive and negative) was 5% BC, suggesting that this dose is more able to specifically modify the microbiota and gene expression. We observed very few significant correlations between gene expression and relative abundance of bacterial pathways; IL-10 gene expression was the one with the highest number of significant negative correlations with several bacterial pathways, and this is in agreement with the low expression of cytokine in our samples and the absence of its gene expression modulation both in the small and in the large intestine.

## 5. Conclusions

Bovine colostrum contains several active molecules that have been demonstrated to affect the modulation of the microbiota, to support the immune response, and to provide high-energy and quality nutrients in humans, rodents, pigs, dogs, and cats. Only a few studies from our group have investigated the effects of BC supplementation in rabbits. At a 5% concentration, colostrum significantly upregulated IL-8 and TGF-β in the jejunum, suggesting a potential role in promoting gut homeostasis. It modulated the gut–vascular barrier, primarily in the jejunum, influencing the Wnt/β-catenin pathway. However, effects varied in the colon. Colostrum had a localized impact on gut wall gene expression, with minimal influence on mesenteric lymph nodes and vaccine response. Correlation analysis between gene expression and bacterial abundance further highlighted the intricate interplay between BC, gut microbiota, and gene modulation, with the 5% BC supplementation showing more pronounced effects. While BC supplementation influenced antioxidant gene expression in the large intestine, the intricacies of this modulation and its potential implications warrant further investigation. Our results suggest that BC supplementation could be a promising nutraceutical in rabbits, but further investigations should demonstrate its capability to protect animals from intestinal infections or dysbiosis, potentiation of antibody responses after different vaccinations, and beneficial effects on meat quality.

## Figures and Tables

**Figure 1 animals-14-00800-f001:**
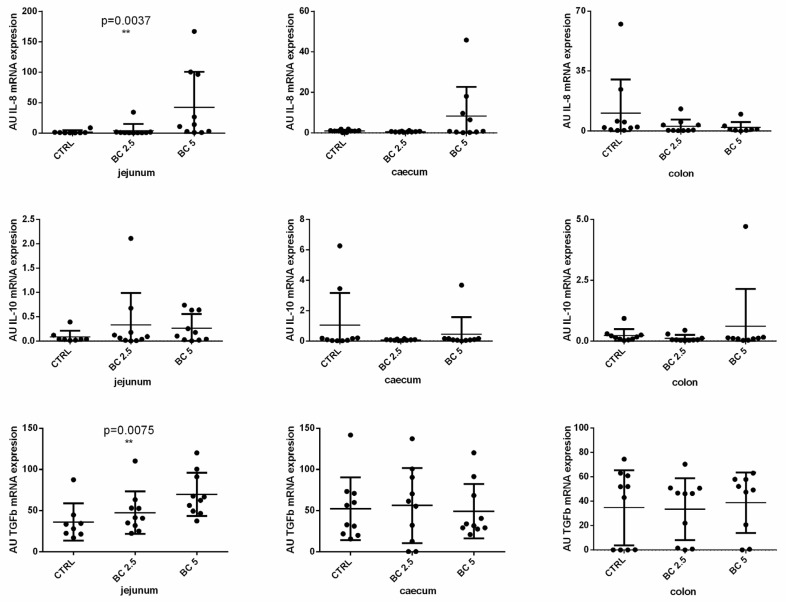
Immune-related molecule gene expression in different tracts of the rabbit gut. IL-8, IL-10, and transforming growth factor beta (TGF-βb) mRNA expression was analyzed by real-time PCR in the jejunum, caecum, and colon tissues of rabbits fed with three different diets. The gene expression level of the target gene was normalized to GAPDH, and the results are presented as arbitrary units (AU). Results are expressed as mean ± SD in all the panels, with ** *p* < 0.01.

**Figure 2 animals-14-00800-f002:**
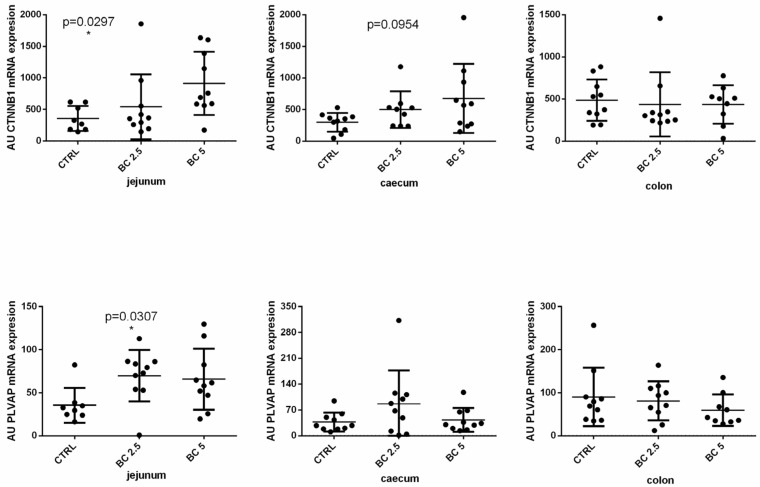
Gut–vascular barrier-related molecule gene expression in different tracts of the rabbit gut. βB-catenin (CTNN-Β1) and plasmalemmal vesicle-associated protein-1 (PLVAP/PV1) mRNA expression was analyzed by real-time PCR in the jejunum, caecum, and colon tissues of rabbits fed with three different diets. The gene expression level of the target gene was normalized to GAPDH, and the results are presented as arbitrary units (AU). Results are expressed as mean ± SD in all the panels, with * *p* < 0.05.

**Figure 3 animals-14-00800-f003:**
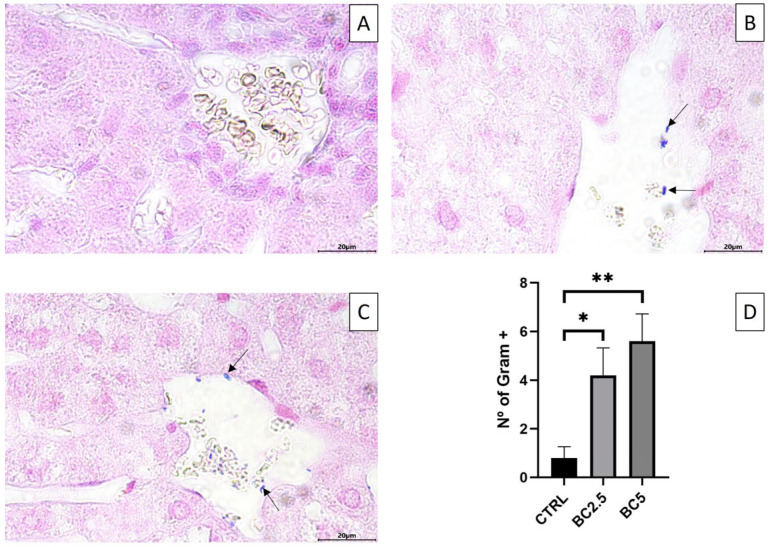
Liver gram staining. Bacteria in the liver’s central veins (arrows) of (**A**) control; (**B**) BC 2.5%; (**C**) BC 5%, scale bar 20 µm. Bacterial counts are represented in (**D**) with * *p* < 0.05; ** *p* < 0.01. Values are expressed as mean ± S.E.M.

**Figure 4 animals-14-00800-f004:**
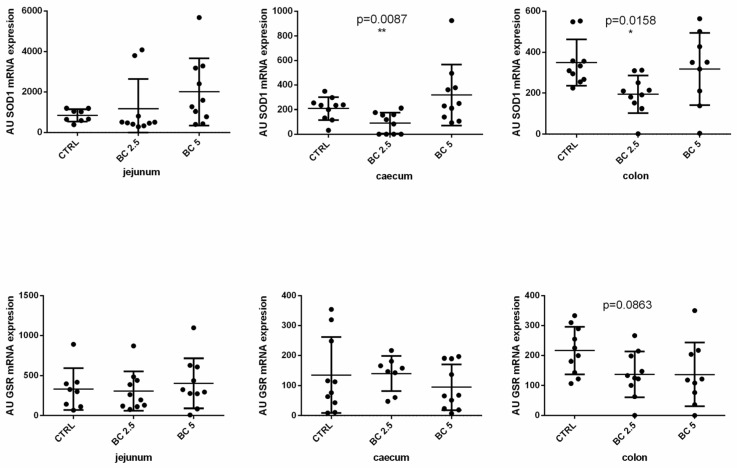
Antioxidant-related molecule gene expression in different tracts of the rabbit gut. Superoxide dismutase (SOD1) and glutathione reductase (GSR) mRNA expression was analyzed by real-time PCR in the jejunum, caecum, and colon tissues of rabbits fed with three different diets. The gene expression level of the target gene was normalized to GAPDH, and the results are presented as arbitrary units (AU). Results are expressed as mean ± SD in all the panels, with * *p* < 0.05; ** *p* < 0.01.

**Figure 5 animals-14-00800-f005:**
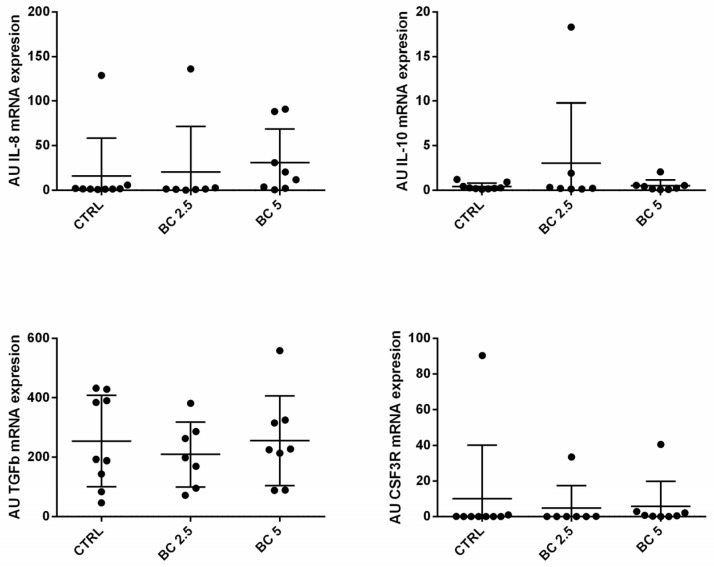
Immune-related molecule gene expression in different mesenteric lymph nodes of rabbits. IL-8, IL-10, transforming growth factor beta (TGF-βb), and Colony Stimulating Factor 3 Receptor (CSF3R) mRNA expression was analyzed by real-time PCR in the jejunum, caecum, and colon tissues of rabbits fed with three different diets. The gene expression level of the target gene was normalized to GAPDH, and the results are presented as arbitrary units (AU). Results are expressed as mean ± SD in all the panels.

**Figure 6 animals-14-00800-f006:**
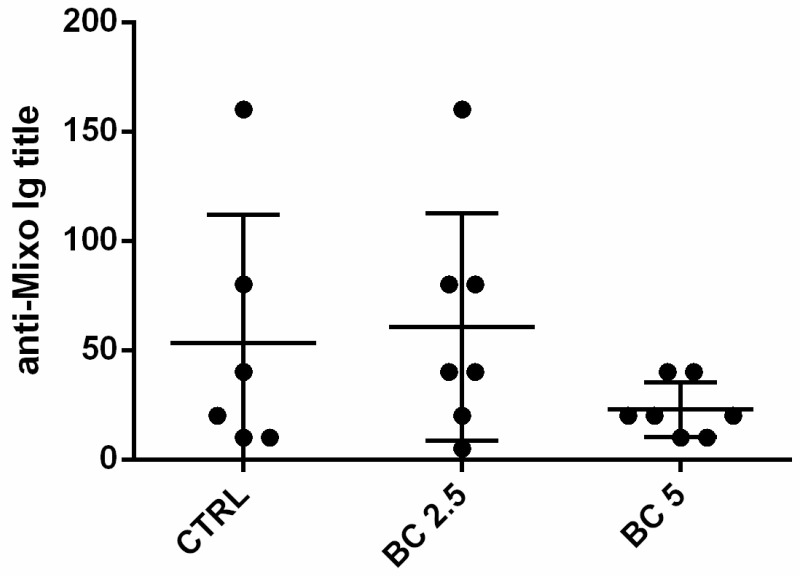
Anti-myxo antibody titration after vaccination in rabbits fed with three different diets. Anti-myxomatosis antibodies were detected in the serum of three groups of rabbits fed with different diets, 28 days after vaccination with CUNIVAX Myxoma.

**Figure 7 animals-14-00800-f007:**
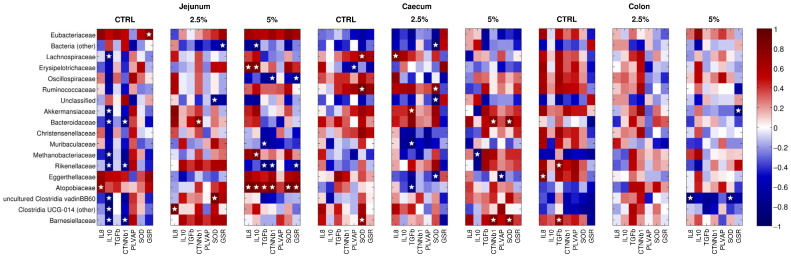
Correlations between gene expression and relative abundance of microbiota families. Heatmaps of the Spearman’s correlation coefficient between the expression of the selected genes and the relative abundances of the main bacterial families (rel. ab > 1% on average), divided by tissue and diet. Blue color indicates a negative correlation, whereas red indicates a positive one; white is correlation = 0. White stars indicate significant (*p* < 0.05) correlations.

**Figure 8 animals-14-00800-f008:**
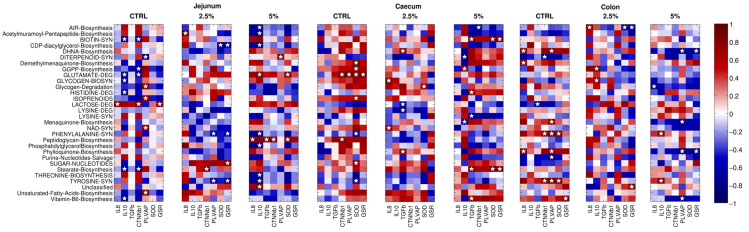
Correlations between gene expression and relative abundance of microbiota pathways. Heatmaps of the Spearman’s correlation coefficient between the expression of the selected genes and the abundances of the bacterial pathways predicted by PICRUST2 (grouped at level 4), divided by tissue and diet. Among all the pathways, only those found as differentially abundant with the diet are considered. Blue color indicates a negative correlation, whereas red indicates a positive one; white is correlation = 0. White stars indicate significant (*p* < 0.05) correlations.

**Table 1 animals-14-00800-t001:** Oligonucleotide primer sequences for SYBR green quantitative RT-polymerase chain reaction amplification.

Gene	Protein	Sequence	Gene Bank GI Number
ACT-β	β-actin	F: ACATGGAGAAGATCTGGCAC	GI:100009272
		R: GCGTGTTGAACGTCTCGAAC	
IL-8	interleukin-8	F: CGGAAGAAACCACTTCGCCT	GI:100009129
		R: ATCCGTGTCAGAACTGCAGC	
IL-10	interleukin-10	F: CCTTTGGCAGGGTGAAGACT	GI:100008701
		R: TCATCTCCGACAAGGCTTGG	
TGF-β	transforming growth factor beta-1 proprotein	F: GCAGAGTGGTTGTCCTTCGA	GI:100008645
		R: CCGGAACTGATCCCGTTGAT	
PLVAP	plasmalemma vesicle-associated protein	F: AAGGACGCCATCATGCAGAT	GI:100343789
		R: TCGAGATGATGGTGGCCATG	
CTNN-β1	β-catenin	F: AGCAGGGCTTTTCTCAGTCC	GI:100125985
		R: ACCCTCTGAGCTCGAGTCAT	
SOD1	CuZn superoxide dismutase	F: CACTTCGAGCAGAAGGGAAC	GI:100009313
		R: CGTGCCTCTCTTCATCCTTC	
GSR	glutathione-disulfide reductase	F: ACGTGAGTCGCCTGAATACC	GI:100337794
		R: GATCTGGCTCTCGTGAGGAC	
CSF3R	granulocyte colony-stimulating factor 3 receptor	F: CGGCCAGTGTGTATCATGTC	GI:100342746
		R: GGTCCCACAGAGGCATAAGA	

## Data Availability

Data will be made available upon reasonable request.

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
