# Peer review of "Bovine Colostrum Supplementation in Rabbit Diet Modulates Gene Expression of Cytokines, Gut–Vascular Barrier, and Red-Ox-Related Molecules in the Gut Wall"

_animals, 2024, doi:10.3390/ani14050800_

Round 1

Reviewer 1 Report

Comments and Suggestions for Authors

Good work, but needs improvement

Introduction 

What is considered high quality BC? Explain

The introduction should indicate the relationship that might exist between the rabbit digestive system and compatibility in complementing BC, justifying its use.

Animals and sample collection

The breed of the cows must be indicated, as well as their origin, whether they are randomly selected commercial farms or individual farms.

Indicate cage dimensions

How was the culling carried out?

The conclusions should be clearer, indicating the doses that have been found to be favourable in this research.

Author Response

Reviewer 1

Good work, but needs improvement.

Introduction 

What is considered high quality BC? Explain

We thank the reviewer for the clarification requests. The requested information regarding colostrum specifications has been added (line 152-155): “More specifically, the colostrum has been obtained from multiparous Frisian cows raised on a single farm, and only high-quality colostrum (Brix grade > 22%) has been used. The quality was tested using a digital refractometer (Palm Abbe, Misco)”.

The introduction should indicate the relationship that might exist between the rabbit digestive system and compatibility in complementing BC, justifying its use.

We thank the reviewer for the suggestion. The requested information has been added (line 119-123 and 128-133): “Thanks to the mentioned features and peculiarities, bovine colostrum could serve as an excellent nutraceutical substance to be added to the diet of rabbits, aiming to strengthen their immune defenses, particularly in the intestinal compartment, to ad-dress gastro-enteric pathologies, which are one of the weak points in rabbit farming” and “The use of an animal product in an herbivorous species is not precluded, when it is a candidate for the improvement of the welfare and production quality and quantity of the animals. Moreover, we used rabbit as a translational model for humans, that al-ready make use of BC as a nutraceutical with several benefits whose molecular mech-anism still unknown are the aim of our study”.

Animals and sample collection

The breed of the cows must be indicated, as well as their origin, whether they are randomly selected commercial farms or individual farms.

We thank the reviewer for the suggestion. The requested information regarding colostrum specifications has been added (line 152-153).

Indicate cage dimensions

The cage dimension has been added (line 161)

How was the culling carried out?

The information about animal culling has been added to the document on lines 164-166: “Rabbits were slaughtered at an official slaughterhouse after stunning by electronarcosis and the carcasses were immediately chilled (4°C) until the analysis time”.

The conclusions should be clearer, indicating the doses that have been found to be favourable in this research.

We thank the reviewer for the suggestion, the conclusions have been rewritten to summarize and present the most significant results more effectively.

Reviewer 2 Report

Comments and Suggestions for Authors

In my opinion, manuscript entitled ,,Bovine colostrum supplementation in rabbit diet modulates gene expression of cytokines, gut vascular barrier, and red-ox related molecules in the gut wall” is valuable and deserves to be published in Animals. I have only a few minor seggestions:

- The authors use only one housekeeping gene (actin β), whereas most scientific publications currently use two or even three such genes. In my opinion, the authors should point out this in the discussion as some kind of methodological weakness. By the way, every study has its weaknesses, so let's not be afraid to point them out.

- In Discussion, the authors unnecessarily repeat information previously provided in the Introduction. E.g. ,,Rabbits are largely used in Europe as pets, laboratory and livestock animals. This species is highly susceptible to intestinal infectious diseases...”

- In Conclusions, authors should focus on summarizing the results of their research. That's why I think the first two sentences don't fit here.

Author Response

Reviewer 2

In my opinion, manuscript entitled Bovine colostrum supplementation in rabbit diet modulates gene expression of cytokines, gut vascular barrier, and red-ox related molecules in the gut wall” is valuable and deserves to be published in Animals. I have only a few minor seggestions:

- The authors use only one housekeeping gene (actin β), whereas most scientific publications currently use two or even three such genes. In my opinion, the authors should point out this in the discussion as some kind of methodological weakness. By the way, every study has its weaknesses, so let's not be afraid to point them out.

We agree with the reviewer. Actin-β was the only housekeeping gene used. Actin-β is a cytoskeletal protein, and together with GAPDH is one of the most common reference genes used in bibliography. Actin-β has been suggested a good reference gene in different studies (Peng et al. 2012; Liu et al., 2022). We decided to use Actin-β as a single reference gene because it is well characterized in rabbits and we had to use a single reference gene for technical reasons (same RNA to test with different primer pairs on the same plate).

In any case we added in the M&M section a sentence that highlighted this peculiar weakness at lines 196-201: ”We used a single housekeeping gene (actin b) both for technical reasons (to have on the same plate numerous primer pairs for the same sample) and because that gene was used in several previous publications on rabbit gut, but we are aware that this approach could be a bias. Indeed, the MIQE guidelines suggest to use at least 3 housekeeping genes to normalize the expression of target genes”.

- In Discussion, the authors unnecessarily repeat information previously provided in the Introduction. E.g. ,,Rabbits are largely used in Europe as pets, laboratory and livestock animals. This species is highly susceptible to intestinal infectious diseases...”

We thank the reviewer for the suggestion. The sentence has been removed, and the text has been reworked to make it flow more smoothly (line 465-467).

- In Conclusions, authors should focus on summarizing the results of their research. That's why I think the first two sentences don't fit here.

The conclusions have been rewritten to summarize and present the most significant results more effectively.

Reviewer 3 Report

Comments and Suggestions for Authors

The manuscript shows some points of interest. However, I detected also several criticisms and points which must be improved to allow a well understanding of the results.

L28 and L36: please use laboratory animals instead of experimental animals

L40-41: the experimental design must be well described. I suppose 30 females of 35 days old were divided into the experimental groups.

L54-55: please add a reference to support this assertion.

L65-67: please, add some references to support this statement. I suggest: 10.1017/S1751731110000558 and 10.1016/j.anireprosci.2017.07.014.

L67: if intended for animals, it is better to use feed and not food.

L120-124: in the definition of the aims of the research, it is important to state the rationale of the use of an animal product in the nutrition of an herbivorous like rabbit.

L136: why only females?

L136-141: Please use a table to report the ingredients of the diets. In addition, all the 3 diets used in the trial must be reported in the table of the ingredients. If you add 2.5 or 5 % of an ingredient another one must be decreased.

L142-143: in what way the BC was used? Dehydrated? Describe the method used to obtain the BC added to the diets.

L144-145: what are the methods used for chemical analysis of the diets?

In the results section some data are mandatory: 1) the weight of the animals at least at the end of the trial to calculate the average body weight gain; 2) the feed intake to verify if the inclusion of BC can affect the palatability of the feed; 3) the feed conversion ratio. For this kind of data, it would be better to compare the means by using the orthogonal contrast analysis (linear and quadratic).

L441-447: this is a repetition of the introduction. Delete.

Author Response

Reviewer 3

The manuscript shows some points of interest. However, I detected also several criticisms and points which must be improved to allow a well understanding of the results.

L28 and L36: please use laboratory animals instead of experimental animals

We thank the reviewer for the suggestion. The term “experimental” has been replaced with “laboratory”.

L40-41: the experimental design must be well described. I suppose 30 females of 35 days old were divided into the experimental groups.

We thank the reviewer for the clarification requests, the sentence has been rephrased in a clearer way (line 41-42).

L54-55: please add a reference to support this assertion.

We thank the reviewer for the suggestion. At line 55 the reference [1] has been added.

L65-67: please, add some references to support this statement.

We thank the reviewer for the suggestion. At line 65 the references [8,9] had been added.

L67: if intended for animals, it is better to use feed and not food.

We thank the reviewer for the suggestion, the term feed has been added in the phrase.

L120-124: in the definition of the aims of the research, it is important to state the rationale of the use of an animal product in the nutrition of an herbivorous like rabbit.

We thank the reviewer for the suggestion. We added a sentence to state the rationale of the use of an animal product in the feed of an herbivorous at line 128-133. Colostrum is a type of food that all the mammals experience in their life, herbivorous, omnivorous and carnivores. The use of bovine colostrum in rabbits means to use a type of food similar to that little rabbits already tasted. Moreover, bovine colostrum has several important qualities already mentioned in the introduction. This manuscript aimed to test the nutraceutical effect of BC in rabbit also under a translational point of view, using rabbit as model for humans. In conclusion there are not preclusion in the administration of animal origin product as feed to herbivorous, unless they are cause of healthy or production problems. Fish meal, meat meal, blood, fish oil etc… are commonly used in the diet of herbivorous production animals. Animal meal were banned from ruminant diet only for BSE emergency.

L136: why only females?

L136 à 146: We avoid the use of males to standardize the samples. Introducing different sexes would have led to the introduction of an additional variable to be evaluated, which would have required a larger number of animals to be sacrificed in order to make a comparison with adequate statistical power.  

L136-141: Please use a table to report the ingredients of the diets. In addition, all the 3 diets used in the trial must be reported in the table of the ingredients. If you add 2.5 or 5 % of an ingredient another one must be decreased.

We appreciated the reviewer's clarification requests. The chemical composition and formulation of the diet are already described and published in a previous paper (Serra et al 2023) that we cited at line 160.

L142-143: in what way the BC was used? Dehydrated? Describe the method used to obtain the BC added to the diets.

L152-155: The description of how the BC was obtained has been added. In the next lines we added that the colostrum was lyophilized and then added to the feed before it was pelleted (lines 158-159).

L144-145: what are the methods used for chemical analysis of the diets?

Dry matter analysis of diets was performed after oven drying the samples at 105°C for 16 h (method 934.01; AOAC International). Dried samples were ground and analyzed for ash using Official Method 924.05 (AOAC Internationa). Ether extract (EE) was determined method 920.39 (AOAC International). Crude protein determined by the Kjeldahl method (method 990.03; AOAC International). NDF, ADF, and ADL were determined using the Van Soest method (method 2002:04; and 973.18; AOAC International).

In the results section some data are mandatory: 1) the weight of the animals at least at the end of the trial to calculate the average body weight gain; 2) the feed intake to verify if the inclusion of BC can affect the palatability of the feed; 3) the feed conversion ratio. For this kind of data, it would be better to compare the means by using orthogonal contrast analysis (linear and quadratic).

We added some of the data required such as the body weight at the end of the trial, in the Results section at line 459-462 and M&M section at line 268-270. All the other data (average body weight gain, the feed intake and the feed conversion ratio) will be included in another manuscript on the effect of the BC on the meat quality and meat production. However, the feed was highly palatable, indeed was completely consumed.

L441-447: this is a repetition of the introduction. Delete.

We are sorry, but we didn’t find at the indicated lines any repetition.

Reviewer 4 Report

Comments and Suggestions for Authors

There are points/values on figures 1, 2, 4, 5 which do not belong to the normal distribution set. They affect the whole statistics, and in this way results and discussion.

PLVAP or PVLAP - only one spelling is correct.

Abbreviations are not well desribed in the text.

Comments on the Quality of English Language

Spelling of PLVAP

Author Response

Reviewer 4

There are points/values on figures 1, 2, 4, 5 which do not belong to the normal distribution set. They affect the whole statistics, and in this way results and discussion.

We thank the reviewer for the comment. We agree with the reviewer and we better clarify in the statistics paragraph of the materials and methods section (line 251-254), that after conducting the Shapiro-Wilk normality test on the data, we applied a suitable test parametric for normally distributed data and non-parametric test for not-normally distributed data (ANOVA and Krustal-Wallis test respectively).

PLVAP or PVLAP - only one spelling is correct.

Thank you for the suggestion: the correct spelling is PLVAP (plasmalemma vesicle associated protein). We have replaced the wrong spelling all along the manuscript.

Abbreviations are not well desribed in the text.

We thank the reviewer for the clarification, all the abbreviation has been described.

Round 2

Reviewer 4 Report

Comments and Suggestions for Authors

Only one reference from 1989.

There are plenty of studies on the use of colostrum as the dietary supplement in 1970's.

istorical

Author Response

Reviewer 4

Only one reference from 1989.

There are plenty of studies on the use of colostrum as the dietary supplement in 1970's

We thank the reviewer for the suggestions, we added two more recent publications (n 13 and 14) about the use of bovine colostrum as a dietary supplement.

  1. Sienkiewicz, M.; SzymaÅ„ska, P.; Fichna, J. Supplementation of Bovine Colostrum in Inflammatory Bowel Disease: Benefits and Contraindications. Adv Nutr 2021, 12, 533–545, doi:10.1093/ADVANCES/NMAA120.
  2. Chandwe, K.; Kelly, P. Colostrum Therapy for Human Gastrointestinal Health and Disease. Nutrients 2021, 13, doi:10.3390/NU13061956.
